# *Mark My Words*: Repurposing LLMs for Specialized Domains via Ability Tokens

## Abstract

Large Language Models (LLMs) have demonstrated remarkable proficiency in natural language understanding and generation. However, their capabilities wane in highly specialized domains, such as biomedical sciences, which are sparsely represented in the pretraining corpus. In this work, we explore how to repurpose general LMs as specialized task solvers. We introduce a novel and systematic framework for adding markup-style language extensions (which we term *"ability tokens"*) to pretrained LMs. These tokens are learned embeddings appended to the LM's embedding matrix, preserving the pretrained weights and the model's original capabilities. We introduce two types of ability tokens: *domain markers*, which delimit and aid in the processing of specialized inputs (e.g., molecular formulas), and *functional tokens*, which guide the model on how to leverage these inputs to solve specific tasks (e.g., predicting molecule properties). During inference, these tokens are inserted into the input text to wrap specialized information and provide problem context. Experimental results show that (i) our markup extensions significantly boost performance in various specialized domains, such as protein and molecular property prediction, matching and outperforming expert models specifically tailored to these tasks, and (ii) we can learn the ability tokens separately and combine them in a modular fashion, achieving zero-shot generalization to unseen tasks. Overall, our framework offers a promising method to enhance LMs with domain-specific knowledge while maintaining their general capacities.

## 1 Introduction

Language Models (LMs) can effectively generate and process text across a wide range of topics due to their general-purpose design. Nevertheless, their performance is reduced in specialized domains that are not well represented in the pretraining corpora. Such specialized domains can encompass areas expressed in natural language but with non-standard token distributions, such as Legal English, all the way to non-linguistic, symbolic representations, such as DNA sequences and chemical formulas. For example, LMs have demonstrated notable deficiencies in processing and interpreting protein sequences (Walker et al., 2023) and SMILES strings (Guo et al., 2023), a specialized textual representation of chemical structure, hampering their performance in various biomedical tasks.

Yet specialized domains are precisely those where LMs could have a larger impact, as observed in practical fields like disease diagnosis and drug discovery (Wang et al., 2023; Vinod et al., 2023). Since training large-scale specialized models from scratch is often infeasible due to scarce in-domain data, recent work has proposed to fine-tune existing LMs for specialized domains. Nonetheless, these approaches compromise the model's general abilities (Singhal et al., 2022; Lewkowycz et al., 2022). Other methods such as in-context learning are simply less effective (Hou & Ji, 2023) and often non-robust (Lu et al., 2022b). We thus ask the question: can we repurpose pretrained LMs for specialized domains in a systematic way that not only achieves effectiveness and data-efficiency but also preserves their general linguistic and reasoning capabilities?

Our proposed solution is to introduce a markup-language-like extension that enables domain- and task-specific behavior of pretrained LMs by inserting special *ability tokens* into the input text (Figure 1). These tokens are parametrized as continuous vectors added to the model's existing embeddings, so the pretrained weights remain intact and the model's general language capabilities are retained. To enhance the LM's performance in specialized domains, we pinpoint two key abilities

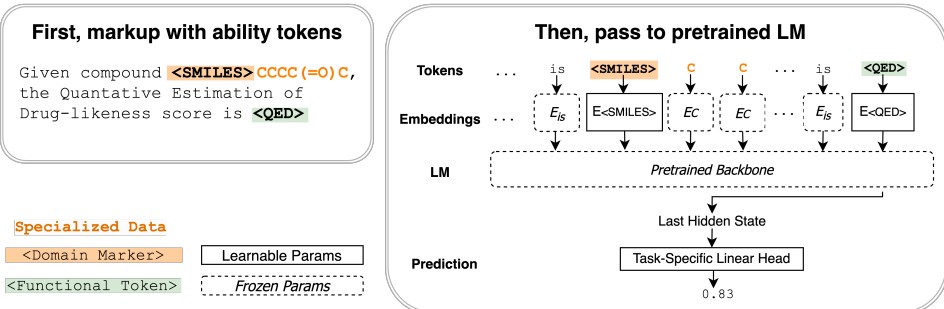

Figure 1: Our method adds domain markers and functional tokens to the input prompt, which are mapped to specially trained embeddings appended to the model's frozen embedding layer. The model's last hidden state is passed to a task-specific head to generate predictions of the desired type (token index, scalar value, etc).

it should have: (i) identifying domain-specific data and processing it accordingly, and (ii) correctly using the data to solve a task. Therefore, we design two types of special tokens:

- **Domain markers** are used to signal the start of specialized representations and prime the model to better process them. They are similar to *descriptive markup* elements (e.g., the $ or \[ delimiters in LaTeX), which describe the nature of the input they surround.
- **Functional tokens** are appended to the input to represent specific tasks and instruct the model on how to solve them. They correspond to *procedural markup* elements (e.g., the \maketitle command in LaTeX), which provide instructions for programs to how to use the text inputs.

In addition, we notice that many tasks in specialized domains involve predicting outputs beyond the next probable word, such as scalar values for regression, probability vectors for classification, or graphs for structured prediction. In response, we augment the functional tokens with task-specific output heads that can be used to replace the language modeling head. This allows us, for example, to directly generate a scalar-valued drug-likeliness score for a chemical compound instead of having the model generate decimals one digit at a time (see Figure 1 and the experiments in Section 4.2), which yields more accurate and targeted predictions.

Now, to train the ability tokens so that they fulfill their roles, we leverage the observation that even within specialized domains, unlabeled data (e.g., molecule formulas without meta-data) is easier to obtain than labeled and highly-structured task-specific data (e.g., molecule formulas paired with names and physical properties). Thus, the generic and task-agnostic domain markers can be learned predominantly using the former, while the task-specific functional markers require the latter. Concretely, we propose a three-stage training protocol for our markup extension:

- **Stage 1**: domain markers are trained for next-token prediction using unlabeled data.
- **Stage 2**: single-domain functional tokens are trained using supervised single-domain data, with the learned domain markers integrated into the data. We explicitly allow the domain markers to be updated at this stage, so they are enriched with relevant domain knowledge.
- **Stage 3**: multi-domain functional tokens are trained using multi-domain supervised data, building upon the enriched markers from previous stages.

Following the above procedure, we can incrementally learn and add new tokens to our markup system, which allows greater flexibility in expanding the LM's capabilities.

We validate the effectiveness of our method on multilingual and biomedical domains using the open-sourced LLaMA models (Touvron et al., 2023). Functionally, we show that the domain markers on their own can act as effective context switchers, enabling transitions between different domains and tasks (*modularity*). Moreover, a single functional token can be combined with different markers to represent different problems, which allows zero-shot generalization to previously unseen tasks (*compositionality*). Empirically, we demonstrate that our markup system can repurpose the LLaMA model into (i) a cross-lingual translator capable of competing with specialized multilingual models; and (ii) a domain expert in therapeutic science, achieving state-of-the-art results on multiple drug-discovery tasks from the Therapeutics Data Commons benchmark (Huang et al., 2021). Given its adaptable, model-agnostic, and plug-and-play nature, our method is poised to retain its utility irrespective of future advancements in LLM capabilities.

## 2 RELATED WORK

### 2.1 DOMAIN ADAPTATION & CONTROLLABLE GENERATION FOR LLMS

Our work aims to adapt LLMs pretrained on general-purpose data to specialized domains with both efficacy and efficiency. Although a similar goal motivates previous work in language modeling (Keskar et al., 2019; Dathathri et al., 2019; Chan et al., 2020; Zhang et al., 2022), these methods mainly consider domains defined in terms of content topics (e.g., business, sports, music), literary forms (e.g., poetry, story), or linguistic styles (e.g., polarity, formality). Different from them, we expand beyond these natural language domains into non-linguistic representations, such as molecule formulas and protein sequences. We also generalize to more complex problems where the target outputs consist of a mixture of natural language and non-text elements, such as scalar-valued scores.

To achieve controllable generation within specific domains, prior work has explored training conditioners from scratch (Keskar et al., 2019) or making use of pretrained models as we do. In the latter case, methods based on hard prompts (Zhang & Song, 2022), soft prompts (Yang et al., 2022), and prefix-tuning (Yu et al., 2021; Qian et al., 2022) have been developed. Although these methods also modify the inputs to achieve adaptation, our work differs from theirs in three key aspects. First, structure-wise, conventional prompt-based techniques are restricted to prefixing the input with either hand-crafted or learned prompts, whereas we insert special tokens into different places of the input text for different purposes (Figure 1). Second, our ability tokens have a hierarchical structure (see Section 3.2 and Figure 2 later), with each type of tokens building upon the lower-level ones, while existing prompts all have flat structures. Lastly, the modularity of our markup system allows us compose different ability tokens together and generalize to unseen tasks, which we show in Section 4. In contrast, previous approaches need to learn a distinct set of parameters for each new task.

### 2.2 TASK COMPRESSION & DECOMPOSITION

A key insight behind our functional tokens is to compress a task into a single special prompt. Several recent works have explored similar ideas of task or instruction abstraction. For instance, Mu et al. (2023) studies how the LM itself can summarize an user-supplied instruction into a reusable soft prompt for computational efficiency. Different from this work, we do not require explicit text instructions but learn the functional tokens implicitly from the data. Shao et al. (2023) represents each NLP task in the T0 benchmark (Sanh et al., 2021), such as multi-choice question answering and sentiment analysis, as a combination of discrete latent codes and examine shared codes across various tasks. As opposed to conducting post-hoc analyses, we separate the task itself (function) from the input properties (domain) during the learning process using two types of tokens. This gives us better control in terms of task decomposition.

### 2.3 MODEL DEVELOPMENT FOR NON-LINGUISTIC DOMAINS

Extensive efforts have been devoted to developing domain-specific large models for specialized tasks such as coding (Ahmad et al., 2020), mathematical reasoning (Zong & Krishnamachari, 2023), and protein structure prediction (Jumper et al., 2021). Yet these models typically have custom architectures and are trained from scratch, thus requiring large amounts of in-domain data and extensive compute. In contrast, here we seek a method that can leverage large pretrained models and requires minimal in-domain data, allowing for a faster on-demand adaptation by non-domain-experts.

In similar vein, a very recent line of work proposes to adapt pretrained LMs to various non-linguistic fields such as physical and biological sciences (Singhal et al., 2022; Vinod et al., 2023; Shen et al., 2023), tabular data (Dinh et al., 2022), and quantitative reasoning (Lewkowycz et al., 2022). These methods require either fine-tuning the entire model on the data embeddings (Lu et al., 2022a; Shen et al., 2023) or manually converting domain-specific data into text, e.g., Dinh et al. (2022) describes each row in a tabular dataset via words. In contrast, here we learn continuous embeddings that condition the model with domain-specific information, which is more parameter-efficient than fine-tuning and more effective than text conversion (see experimental results in Section 4).

## 3 METHODS

In this section, we outline our approach to building a markup-style extension that tailors pretrained LMs to specialized domains. We begin with an overview of the method, detailing the intuition behind its design (Section 3.1). Then, we introduce two types of ability tokens, domain markers and functional tokens, as well as a three-stage protocol for training them (Section 3.2).

### 3.1 MOTIVATION: A SYSTEMATIC FRAMEWORK TO ADAPT LMS TO SPECIALIZED DOMAINS

We consider the problem of repurposing a pretrained general-purpose LM to solve tasks in specialized domains, i.e., domains that are underrepresented in the model's pretraining corpus but for which we have limited data available for adaptation. We are especially interested in technical and rapidly evolving domains, such as those at the frontiers of science, where new previously unseen data is constantly being generated, and knowledge is represented through a mixture of linguistic and non-linguistic data. These two characteristics (namely, unobserved knowledge and heterogeneous data) make pretrained LMs ill-suited for solving specialized problems without further modification.

Existing adaptation techniques are insufficient for such specialized domains. In particular:

- **Full fine-tuning** is computationally costly for most LMs and requires substantial labeled data.
- **Adaptation by hard-prompting**, e.g., by providing instructions or domain-specific information via demonstrations (Brown et al., 2020), does not require gradient updates and is thus more efficient. However, it often requires cumbersome trial-and-error prompt crafting and can be highly sensitive to minor changes in the format of the prompt (Lu et al., 2022b). Meanwhile, the amount of domain information included in the prompt is limited by the LM's context length.
- **Parameter-efficient fine-tuning methods** such as LoRA (Hu et al., 2021) and prefix-tuning (Li & Liang, 2021) offer a middle ground between the above approaches, modifying fewer parameters and less limited by context length. However, they can typically adapt to a single downstream task at a time and learn distinct sets of parameters even for tasks that share knowledge (e.g., in the same domain). Furthermore, they are not equipped with the ability to handle non-linguistic information.

The framework we propose combines the benefits of these approaches while overcoming their limitations. We enhance the model's vocabulary with small sets of parameters that are pertinent to both individual tasks and broader domains, aiming for more systematic, modular, and 'persistent' adaptation than current prompt- or prefix-optimization techniques. At the core of our approach lies the design of a family of markup-style "ability tokens" that can be learned to condense domain- and task-relevant information. Inspired by prompt tuning (Lester et al., 2021a) and extensions thereof (e.g., Li & Liang, 2021), we represent the ability tokens as continuous embeddings which can be added to the LM's embedding matrix once learned. During inference, the ability tokens are inserted to the input text to provide useful contextual knowledge (Figure 1).

### 3.2 ABILITY TOKENS: DESIGN, PARAMETRIZATION, AND TRAINING

Driven by the goal of distinguishing between general domain information and task-specific instructions, we introduce two distinct categories of ability tokens: *domain markers* and *functional tokens*. The domain markers delimit the occurrence of domain-specific data (e.g., French or protein sequences) in the input and encode domain-level information that can be relevant to multiple downstream tasks. Functional tokens, in turn, guide the model to use these domain inputs to solve specific tasks (e.g., translation or protein property prediction) by encoding task semantics. These two types of tokens are reminiscent of descriptive and procedural markup, respectively (Section 1).

Both types of tokens are parametrized in a similar fashion. Given a LM with pretrained vocabulary $\mathbb{V}$ and embedding dimension $d$, we instantiate each ability token as a learnable parameter in $\mathbb{R}^{p \times d}$, where $p$ denotes the length of the ability token. Here, $p$ is a tunable hyperparameter so the ability tokens can have length greater than 1. Also note that there is no word in the model's dictionary corresponding to an ability token, and the model cannot output an ability token. We initialize the parameters using the model's average embedding, $\hat{v} = \frac{1}{|\mathbb{V}|} \sum_{v \in \mathbb{V}} v$. More specifically, we first scale $\hat{v}$ by $\frac{1}{|\mathbb{V}| \|\hat{v}\|} \sum_{v \in \mathbb{V}} \|v\|$ to match its norm with the average norm of the ordinary token embeddings in $V$. Then, we stack $p$ copies of the re-scaled $\hat{v}$ to initialize an ability token.

Given that the two types of tokens have distinct natures and roles, we leverage different types of data to train them. Domain markers, which are more general and task-agnostic, can be trained using unlabeled data from a single specialized domain (e.g., amino acid sequences for representing proteins). Such unsupervised data is often available in relatively large amounts. Functional tokens, on the other hand, require labeled data specific to the task they encode (e.g., predicting protein properties based on its sequence), which is scarcer. Functions that involve multiple domains (e.g., predicting the

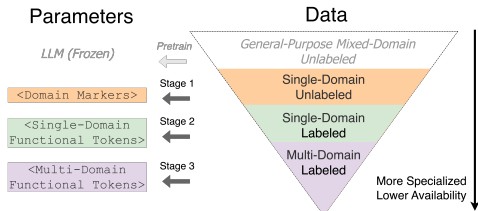

Figure 2: We observe a hierarchy of specialized domain data and correspondingly design a three-stage training protocol for the special tokens.

binding affinity between a protein and a drug) require multi-domain labeled data, the scarcest kind.

Therefore, we design a hierarchical training protocol that develops ability tokens progressively, from general to specialized (Figure 2). Importantly, the training of functional tokens leverages domain information already encoded in the markers, making it more sample-efficient. The markers can also be fine-tuned during later stages to incorporate more domain knowledge, which we will discuss later. In general, our modular and systematic way of conditioning the LMs with special tokens not only enables more granular control over the LM but also allows composing multiple tokens for solving different and potentially unseen tasks (we will show this in Section 4.1). Meanwhile, new tokens can easily be added to the system, regardless of whether they depend on existing tokens or not. In the following sections, we discuss the training and usage for each type of ability tokens in detail.

### 3.2.1 STAGE 1: SINGLE-DOMAIN DOMAIN MARKERS

**Usage.** Our intuition is that markers for different domains (e.g., French, amino acids, molecule formulas) should summarize domain-relevant knowledge. Then, inserting them to the input should provide the context more efficiently—we can shift the LM's conditional distribution at the moment a marker appears, rather than letting the model determine the context itself after processing the input data. To this end, we insert the marker right before the specialized tokens. Denote the marker as $M \in \mathbb{R}^{p \times d}$. Given a specialized representation $\{x_1, \cdots, x_n\}$ whose embedding matrix is $X_e \in \mathbb{R}^{n \times d}$, we prepend $M$ to $X_e$ so that the embedded input used to query the LM is $[M; X_e] \in \mathbb{R}^{(p+n) \times d}$.

**Training.** We train the markers using in-domain demonstrations via self-supervised next-token prediction. That is, $M$ is learned by optimizing $l_M := \sum_{t \in [n]} \mathbb{P}(x_{t+1} | [M; x_{1:t}])$. We use the standard autoregressive mask during training and inference. At later stages, the markers can be further fine-tuned ("enriched") along with the functional tokens to encapsulate domain knowledge. To ensure the markers maintain their role during enrichment, we also compute and optimize $l_M$ using the specialized representations.

### 3.2.2 STAGE 2 & 3: TASK-SPECIFIC FUNCTIONAL TOKENS

We differentiate between single-domain functions (requiring one marker) and multi-domain functions (requiring multiple markers). This is because we allow the markers to be enriched in the former case, but in the latter case, they are frozen to avoid being polluted by out-of-domain information.

**Usage.** The goal of functional tokens is to encode specific tasks in specialized domains. However, rather than simply summarizing some user-provided instructions as is done in prior work (Mu et al., 2023), our functional tokens learn the essence of the tasks entirely from the data. To achieve this, we place them at the end of the input so that they can attend to both the specialized representations and the domain markers used to provide extra context. Formally, given a labeled dataset $\{(X, y)\}$, we first preprocess the data by inserting the domain markers $M$ into $X$ immediately before any specialized data. Then, we append the functional token $F$ to the end. Prediction is based on the last hidden state of the LM (corresponding to $F$), and can take several forms (see Section 3.2.3).

**Training.** The functional tokens are learned by directly optimizing the downstream task loss $l_F$. For instance, for text generation, $l_F$ is the standard cross-entropy loss; for regression-based tasks, $l_F$ can be the mean-squared-error between the model output and the label.

### 3.2.3 EXTENDING BEYOND TEXT GENERATION

The ability tokens defined so far adapt the pretrained LMs at the *input level*. Yet, specialized domains (especially scientific ones) often involve tasks whose output is not naturally represented as text, such as scalar-valued regression scores or vector-valued probability distributions for classification. The naive solution is to have the model generate text-based outputs and convert to the desired format afterward. However, this approach is fraught with problems. First, the cross-entropy (CE) loss is not a good proxy for regression losses, such as mean-squared-error (MSE), because the former is agnostic to digit order. For instance, CE("`3.14`", "`3.24`") = CE("`3.14`", "`3.15`") since both pairs of numbers differ by one digit, but $MSE(3.14, 3.24) \neq MSE(3.14, 3.15)$. Additionally, certain problems involve constraints on the output, such as $\|x\|_1 = 1$, which cannot be easily enforced in text-based outputs.

To overcome these limitations and extend the capacity of LMs beyond text generation, we pair the input functional tokens for numerical prediction problems with task-specific output regression heads. Formally, given a task $t$, we initialize $w_t \in R^{d \times d_t}$, where $d_t$ is the desired output dimension (e.g., 1 for univariate regression, $k$ for $k$-class classification). The prediction is obtained by multiplying $w_t$ with the model's last hidden state, and can be scored using any desired loss such as MSE. We show in Section 4 that task-specific output modeling can significantly boost the downstream performance.

## 4 EXPERIMENTS

In this section, we provide empirical evidence that our markup extension can effectively repurpose LMs for specialized domains. We start with a multilingual translation example to showcase our method's modularity and compositionality. Then, we dive into two non-linguistic domains, protein and drug modeling, and demonstrate that our method is competitive with task-specific models on various challenging tasks. In all tasks, we leverage the pretrained LLaMA-7B model (Touvron et al., 2023) with embedding dimension $d = 4096$. We set the token length $p = 10$, resulting in a total of 40960 trainable parameters for each ability token. We run our experiments on a single NVIDIA A100 GPU. More experiment details can be found in Appendix A.1.

### 4.1 GETTING STARTED: A MODULAR MULTILINGUAL TRANSLATION SYSTEM

In our first set of experiments, we consider the problem of translating across various languages. Here, the domain markers are used to delimit the text in different languages, and we train a shared functional token ⟨`Translate`⟩ to encode the translation task. Our goal is to verify (i) if the markers can correctly extract the domain information from the data (*modularity*); and (ii) the learned functional token can generalize to unseen domains and translation pairs (*compositionality*).

Using the OPUS-100 datasets (Zhang et al., 2020), we first learn 8 different language markers, namely ⟨`EN`⟩, ⟨`FR`⟩, ⟨`RU`⟩, ⟨`DE`⟩, ⟨`IT`⟩, ⟨`EL`⟩, ⟨`ES`⟩, and ⟨`PT`⟩, each trained separately with the next-token prediction objective described in Section 3.2.1. Then, we take 5 paired datasets, namely EN-FR, EN-RU, EN-DE, EN-IT, and EN-EL, combine them, and format each example as follows:

> Input:  ⟨source_lang marker⟩ source_text
> Output: ⟨target_lang marker⟩ ⟨Translate⟩ target_text

We train ⟨`Translate`⟩ on the combined dataset using the cross-entropy loss over `target_text`. Note that the training data for this functional token involves only 6 out of the 8 languages (excluding ES and PT). That is, there are 6 domains and 5 language pairs "seen" by ⟨`Translate`⟩.

Our evaluation is done on a subset of the Flores-101 devtest set (Goyal et al., 2021). We compute the sentence-level SentencePiece BLEU and compare with the following LM baselines: the translation-task-specific M2M (Fan et al., 2020), the more general multilingual pretrained XGLM (Lin et al., 2021) and BLOOM (Scao et al., 2022), and the most general GPT-3 (Brown et al., 2020). Note that our primary goal is not to beat the state-of-the-art, since LLaMA is mainly trained on English rather than multilingual data. Instead, we are interested in studying whether we can learn ability tokens that encode the multilingual translation task *from scratch* without extra context (note that our prompt consists only of the ability tokens and the text data, but not instructions). The results can be found in Table 1 and are summarized below.

Table 1: spBLEU scores on Flores-101 datasets (Goyal et al., 2021), with baseline values taken from Lin et al. (2021); Scao et al. (2022). In general, our method achieves performance comparable to that of the multilingual specialized baselines, and the ability tokens successfully generalize to previously unseen combinations, which proves the compositionality of our framework.

| | | Seen Language Pairs | | | | | | Seen Domains, Unseen Pair | | Unseen Domains | |
| --- | --- | --- | --- | --- | --- | --- | --- | --- | --- | --- | --- |
| | | EN-FR | FR-EN | EN-RU | RU-EN | EN-DE | DE-EN | IT-FR | FR-IT | ES-PT | PT-ES |
| General Pretrained | OURS | 40.5 | **45.7** | 14.2 | 30.1 | 23.0 | **40.5** | 28.8 | 23.9 | 24.6 | 27.8 |
| | GPT-3 | 36.1 | 42.8 | 11.2 | 28.1 | 25.9 | 40.4 | - | - | - | - |
| Specialized Pretrained | XGLM | 36.0 | 40.4 | 24.2 | **30.4** | 27.6 | 38.8 | - | - | - | - |
| | BLOOM | **45.0** | 45.6 | - | - | - | - | 31.4 | 24.0 | **29.1** | **28.1** |
| Supervised | M2M | 42.0 | 37.2 | **27.1** | 27.5 | **32.6** | 35.8 | **34.4** | **28.6** | 28.1 | 26.9 |

**Comparable performance on seen domains.** For seen domains pairs, our method performs on par with the multilingual baselines in general, ranking first in two settings. This shows that our ability tokens can effectively compress domain- and task-relevant information and do so in a modularized way. We also observe that all general-purpose models (ours and GPT-3) are better at generating English than other languages, likely due to its predominance in the pretraining corpus.

**Zero-shot generalization to unseen domains.** We observe that our method yields reasonable performance for the language pair ES↔PT (last columns in Table 1), even though ⟨Translate⟩ has never seen either of the languages before. This shows that our functional tokens can extract the shared property of multiple tasks, and our markup system achieves compositionality. Such zero-shot adaptation ability allows us to reuse learned tokens and solve new problems with ease.

Despite the satisfactory performance, the translation example does not fully showcase the potential of our method, since the task requires only language reasoning, which is an inherent ability of general LMs. In the next section, we tackle much more challenging tasks that require specialized knowledge not present by default in general-purpose models.

## 4.2 REAL-WORLD USE CASES: REPURPOSING LMS FOR SCIENTIFIC DISCOVERY

Now that we are familiar with training and applying the ability tokens, we move on to the main focus of our method—specialized representations beyond language. We consider scientific domains where developing domain-specific large models is costly due to limited data and the need for domain expertise. We focus in particular on proteins, represented by amino acid sequences, and chemical compounds, represented by the simplified molecular-input line-entry system (SMILES) notation.

This section is organized as follows. First, we introduce the ⟨Protein⟩ and ⟨SMILES⟩ markers and present results on two molecular property prediction tasks, where our method outperforms prompt tuning and fine-tuning baselines by a large margin. After that, we study two real-world drug discovery datasets from the TDC benchmark (Huang et al., 2021): drug-drug interaction prediction, where our method obtains state-of-the-art results, and drug-protein binding affinity prediction, where it performs competitively against highly specialized expert models. We also provide an ablation study for evaluating the effect of various design and hyperparameter choices.

### 4.2.1 ⟨PROTEIN⟩, ⟨SMILES⟩ & MOLECULAR PROPERTY PREDICTION

In biomedical research, a protein is usually represented as a string of letters, where either a single- or three-letter code represents an amino acid, ordered from the amino-terminal to the carboxyl-terminal of the protein. Similarly, SMILES is a notation system for describing the structure of chemical compounds, e.g., $CC(=O)NC1=CC=C(C=C1)O$ for Acetaminophen (Tylenol). Previous work has shown that general LMs on their own exhibit poor understanding of these special notations (Walker et al., 2023; Wang et al., 2023). Nonetheless, we show that our markup system allows LMs to process these representations and solve prediction tasks with remarkable precision.

Following our three-stage protocol, we first train the ⟨Protein⟩ and ⟨SMILES⟩ markers for next-token prediction[1] using unlabeled data extracted from Blanchard et al. (2021). Then, we investigate

---

[1] Many LM tokenizers automatically group several characters together for protein/SMILES sequences, introducing tokenization bias. Thus, we manually enforce every character to be a separate token in our workflow.

Table 2: Performance of our markup system vs. baselines on protein descriptor prediction and SMILES QED prediction ($\downarrow$\$\uparrow$: lower\higher is better). All tuning methods use LLaMA-7B. Our method significantly outperforms the others, showing effectiveness for adapting LMs to regression tasks in non-linguistic domains.

|  |  | Descriptor Prediction | | QED Prediction | |
|---|---|---|---|---|---|
|  |  | MSE ($\downarrow$) | Pearson $r$ ($\uparrow$) | MSE ($\downarrow$) | Pearson $r$ ($\uparrow$) |
| Tuning-Based | OURS | **0.005** | **0.717** | **0.008** | **0.918** |
|  | LORA | 0.006 | 0.645 | 0.020 | 0.745 |
|  | PROMPT TUNING | 0.011 | 0.182 | 1.387 | -0.123 |
|  | LINEAR PROBING | 0.041 | 0.067 | 0.012 | 0.883 |
| Tuning-Free | HARD PROMPT | 0.236 | -0.258 | 0.298 | -0.008 |
| Non-Parametric | NEAREST NEIGHBOR | 0.012 | 0.091 | 0.040 | 0.528 |

two single-domain tasks: descriptor prediction for proteins, and Quantative Estimation of Drug-likeness (QED) score prediction for molecules. The ground truth labels are automatically generated using the Python peptides and RDKit libraries, respectively. Since the tasks involve numerical prediction, we pair the functional tokens with regression heads and train them using the MSE loss. We evaluate three sets of baselines: (i) parameter-efficient fine-tuning methods, including LoRA (Hu et al., 2021), prompt tuning (Lester et al., 2021b), and vanilla linear probing; (ii) hard-prompting the model with task instructions; and (iii) a simple non-parametric nearest-neighbor algorithm that looks for similar training data points and leverages their labels to make a prediction. More details about the tasks, baselines, and implementation can be found in Appendix A.2.

We report test-set MSE and Pearson's correlation coefficient in Table 2. Notably, our method obtains the lowest error rates and the highest correlation scores among all baselines on both tasks, which underscores the benefits of explicitly adapting pretrained LMs to non-linguistic domains using our markup system. We also compare the number of learnable parameters for each approach in Appendix A.2.1. Our method is much more parameter-efficient than the second-ranked LoRA, and much more effective than vanilla prompt tuning, which has the same number of learnable parameters as our method. Encouraged by these results, we turn to even more challenging tasks involving real-world multi-instance prediction problems for drug discovery in the next section.

### 4.2.2 DRUG-DRUG INTERACTION & ABLATION STUDIES

The Therapeutics Data Commons (TDC) (Huang et al., 2021) covers a wide range of tasks in therapeutic modalities related to target discovery and activity modeling. We start with the Drug Combination (DC) task that predicts the sensitivity score between a pair of drugs, using the DrugComb_CSS benchmark. To process the data, we first tag the SMILES strings representing the drug compounds with the $\langle$SMILES$\rangle$ token and then append the $\langle$DC$\rangle$ token to the end of the input. We learn the special tokens as well as the task regression head using the MSE loss.

**Achieving state-of-the-art on drug combination.** Following the benchmark, we compute the mean-absolute-error (MAE) on the test set (Table 3). It is worth noting that we outperform not only all LM-based baselines but also the domain-specific expert model (Xia et al., 2018) trained with supervised in-

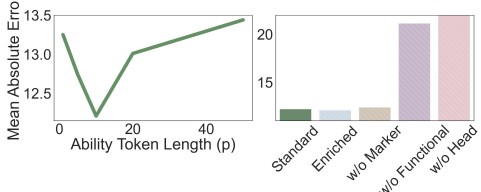

Figure 3: **Left:** MAE (smaller is better) for varying lengths of the $\langle$DC$\rangle$ functional token. As the value of $p$ increases, the test error initially decreases and then starts to rise. The empirically optimal value is $p = 10$. **Right:** MAE for different ablation settings. Enriching domain marker improves performance, whereas removing the ability tokens or regression head impedes performance. For full sets of results, see Appendix A.1.3
.

domain data. This shows the benefit of leveraging LMs for solving specialized tasks, since these pretrained models are both large in scale and equipped with general knowledge. Now, to gain a better understanding of this exceptional performance, we conduct a series of ablation studies on the design choices of our method.

**Effect of token length.** We vary the length of the $\langle$DC$\rangle$ token, taking $p \in \{1, 5, 10, 20, 50\}$. As shown in Figure 3 (left), as $p$ becomes larger, the test error first decreases and then increases. This suggests that, while the added degrees of freedom are initially beneficial, going beyond a certain threshold might result in overfitting to the training data, thus hindering the test-time performance. Consequently, being parameter-efficient does not always imply reduced efficacy.

**Effect of token types & regression head.** Since our markup system consists of three key parts—domain markers, functional tokens, and regression heads—we now dissect the workflow by ablating over the individual components. The barplot in Figure 3 illustrates that all three components are crucial for our method, and removing any of them could result in a decline in performance. In particular, switching from scalar prediction via the regression head to digit generation via the language modeling head results in the largest performance gap, which empirically validates our hypothesis in Section 3.2.3 that next-token prediction is ill-suited for regression tasks. The functional tokens also play an important role given that our evaluation is based on downstream task performance.

**Effect of marker enrichment.** Recall that Section 3.2.1 discusses how the domain markers can be enriched during single-domain task functional token training. We verify that the enrichment is indeed beneficial in Figure 3 (right), where using the ⟨SMILES⟩ marker enriched with QED knowledge has a lower error rate than directly using the non-enriched marker.

### 4.2.3 CROSS-DOMAIN MODELING: BINDING AFFINITY PREDICTION

The Drug Combination task still operates within a single SMILES domain. Hence, to complete our evaluation, we focus on cross-domain modeling in this section. Specifically, the binding affinity measures the activity between a small-molecule drug and a target protein. Traditional methods to gauge the affinities require expensive wet-lab experiments, thus limiting the number of candidate drugs that researchers can search over. However, if we can repurpose LMs for predicting binding affinities, we can not only reduce the pharmaceutical research costs but also enlarge the search space to avoid missing potential candidates.

Table 3: Performance of our markup system vs. baselines on two TDC benchmarks (Huang et al., 2021). We outperform task-specific specialized models on Drug Combination and almost match them on Binding Affinity. We perform significantly better than the other baselines.

| | | Drug Combination | Binding Affinity |
|---|---|---|---|
| | | MAE ($\downarrow$) | Pearson $r$ ($\uparrow$) |
| Tuning-Based | OURS | **12.21** | 0.527 |
| | LORA | 13.46 | 0.125 |
| | PROMPT TUNING | 17.65 | 0.054 |
| | LINEAR PROBING | 24.11 | 0.180 |
| Tuning-Free | HARD PROMPT | 23.39 | 0.017 |
| Specialized | ENSEMBLE MODEL | - | **0.588** |
| | PLM-BASED | - | 0.538 |
| | MLP-BASED | 16.85 | 0.433 |

To see how our method performs on this challenging task, we use TDC's BindingDB_Patent benchmark. The training and test sets are split by the label's patent year to simulate distribution shift. We report the Pearson $r$ in Table 3. Once again, our method outperforms all LM-based baselines, providing evidence that the learned ability tokens are robust under distribution shifts. However, we rank the third among all models submitted to the leaderboard, underperforming the ensemble-based Otter-Knowledge (Lam et al., 2023) and the PLM model (Sledzieski, 2022), all of which are highly specialized methods based on pretrained protein representations. The small performance gap shows the potential of exploiting pretrained LMs for real-world scientific discovery problems. We believe that an interesting future direction of this work is to explore whether scaling from LLaMA-7B to larger models can results in similar or better performance than these specialized methods.

## 5  CONCLUSION AND FUTURE WORK

In this work, we exploit off-the-shelf pretrained LLMs for solving tasks in data-limited specialized domains. We developed a markup-style LM extension that inserts domain- and task-specific ability tokens into the input text and propose a three-stage training protocol to learn these tokens. Our experimental results show that this method improves the LM's prediction quality and allows for more fine-grained control over its behavior. We envision that open-sourcing the learned tokens for different models can be helpful to facilitate research and practical deployment of LLMs in real life.

We identify several future directions based on our work. For example, we can further validate our method in other specialized domains, such as gene function prediction (computational biology) or solving partial differential equations (physics). The idea of enhancing functional tokens with task-specific output heads can be applied to various prediction problems. However, in this study, we mainly focused on regression, leaving the exploration of classification and other structured prediction problems for future research. In terms of computational efficiency, a potential improvement could come from training the ability tokens in large batches, e.g., by concatenating data from different domains together, instead of sequentially as we do here. Lastly, integrating our work with other adaptation paradigms, such as in-context learning, presents an intriguing possibility for exploration.

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

# A  APPENDIX

## A.1  EXPERIMENT DETAILS: OUR METHOD

### A.1.1  GENERAL SETUP

Our experiments leverage the LLaMA-7B (Touvron et al., 2023) model as the backbone. We obtain the checkpoints from here: https://huggingface.co/huggyllama/llama-7b/tree/main. We run our workflow on a single NVIDIA A100 GPU, using FP16 training and evaluation. Below are the hyperparameters used for our experiments (excluding ablation studies).

- Ability token prompt length: 10
- Batch size: 4
- Gradient accumulation: 8
- Optimizer: AdamW
- Learning rate: 1E-4
- Weight decay: 0
- Learning rate scheduler: cosine (with warmup rate 0.03)

### A.1.2  TASK-SPECIFIC SETUP

Here we provide more information on the tasks used in our evaluation.

**Translate**    For training the language markers and ⟨Translate⟩ token, we use the OPUS-100 dataset (Zhang et al., 2020), which is an English-centric multilingual corpus covering 100 languages, i.e., all training pairs include English on either the source or target side. For evaluation, we use a subset of the FLORES-101 dataset (Goyal et al., 2021) and report results for the devtest split. The baseline results are taken from Lin et al. (2021); Scao et al. (2022).

**Descriptor**    peptides is a Python package that computes common descriptors for protein sequences. In our experiments, we first obtain the protein sequences from the binding_affinity dataset (dropping the other features). We then use peptides to compute the average BLOSUM indices for all the amino acids in the peptide and retain the first index as our target label. BLOSUM indices were derived from physicochemical properties that have been subjected to a VARIMAX analysis (Georgiev, 2009).

**QED**    Similar to descriptor, we first collect SMILES sequences from the binding_affinity dataset. Next, we make use of the rdkit.Chem.QED module to compute the quantitative estimation of drug-likeness (QED) score. QED was initially introduced by Bickerton et al. (2012). The empirical rationale of this measure reflects the underlying distribution of molecular properties including molecular weight, logP, topological polar surface area, number of hydrogen bond donors and acceptors, the number of aromatic rings and rotatable bonds, and the presence of unwanted chemical functionalities.

**Drug Combination**    Drug combinations offer exciting new treatment opportunities to increase drug use and efficacy. For instance, simultaneously modulating multiple targets can address the issue of drug resistance seen in cancer treatments. However, experimentally searching the space of possible drug combinations is often infeasible. Thus, developing ML models for this purpose can be highly useful. In our experiments, we focus on predicting the synergy (the deviation of observed drug combination response from expected effects had non-interaction) using the TDC.DrugComb_CSS dataset (Huang et al., 2021). The target value is derived using the relative IC50 values of compounds and the areas under dose-response curves.

**Binding Affinity**    This task requires predicting the interaction activity score between a drug and a target protein using only the compound structural information and protein's amino acid sequence. It is practically meaningful in that identifying high-affinity compounds is the first crucial step for drug discovery. In our experiments, we directly use the BindingDB datasets from TDC's dti_dg_group.

Additionally, we use OPUS-100 to train the language markers and the binding_affinity dataset to train ⟨Protein⟩ and ⟨SMILES⟩. More details about data splits and processing for each task can be found in our supplementary code.

In the following, we summarize how we format the inputs of different tasks to query the language model. In the table, ⟨input⟩ indicates the place where we insert the raw input of a data point, whereas ⟨output⟩ indicates the label (or prediction value) of that data point. The other ⟨⟩'s denote the ability tokens. Note that for regression tasks (whose loss functions are MSE), we take the output hidden states of the functional tokens and pass them to the linear head.

| Task | Train Epochs | Loss | Eval Metric | Input Format |
|---|---|---|---|---|
| Translate | 1 | CE | ROUGE | # # Input:  ⟨src_lang⟩ ⟨input⟩ |
| | | | COMET | # # Output:  ⟨tgt_lang⟩ ⟨Translate⟩ ⟨output⟩ |
| Descriptor | 4 | MSE | MSE | # # Input:  The protein sequence is ⟨Protein⟩ ⟨input⟩ |
| | | | | # # Output:  The descriptor value is ⟨Descriptor⟩ ⟨output⟩ |
| QED | 2 | MSE | MSE | # # Input:  The SMILES of the molecule is ⟨SMILES⟩ ⟨input⟩ |
| | | | | # # Output:  The quantitative estimate of druglikeness is ⟨QED⟩ ⟨output⟩ |
| DC | 2 | MSE | MAE | # # Input:  Drug 1 is ⟨input 0⟩.  Its SMILES is ⟨SMILES⟩ ⟨input 1⟩. |
| | | | | Drug 2 is ⟨input 2⟩.  Its SMILES is ⟨SMILES⟩ ⟨input 3⟩ |
| | | | | # # Output:  The drug combination sensitivity score is ⟨DC⟩ ⟨output⟩ |
| BA | 4 | MSE | Pearson | # # Input:  The protein sequence is ⟨Protein⟩ ⟨input 0⟩. |
| | | | | The SMILES of the drug is ⟨SMILES⟩ ⟨input 1⟩ |
| | | | | # # Output:  The binding affinity is ⟨BA⟩ ⟨output⟩ |

### A.1.3 FULL ABLATION RESULTS

| | Domain Markers | Functional Token | Regression Head | MAE | Pearson $r$ |
|---|---|---|---|---|---|
| Standard | ✓ | ✓ | ✓ | 12.21 | 0.6209 |
| Enriched | ✓ (QED Enriched) | ✓ | ✓ | 12.10 | 0.6259 |
| w/o Marker | | ✓ | ✓ | 12.39 | 0.6079 |
| w/o Functional | ✓ | | ✓ | 21.14 | 0.4617 |
| w/o Head | ✓ | ✓ | | 23.42 | 0.3291 |

## A.2 EXPERIMENT DETAILS: BASELINES

### A.2.1 PARAMETER EFFICIENT FINE-TUNING BASELINES

We use the Hugging Face PEFT library to implement the parameter-efficient fine-tuning baselines, including LoRA (Hu et al., 2021) and prompt tuning (Lester et al., 2021b). For fair comparison, we use the same sets of training hyperparameters (e.g., number of epochs, batch size, learning rate, etc) as our method. Below, we detail method-specific configurations.

- LoRA: rank=8, lora_alpha=16, lora_dropout=0.05

- Prompt tuning: for fair comparison, we set the num_tokens field to the same as the total length of all ability tokens inserted to the input (e.g., 20 for QED and 30 for BA); we use the PromptTuningInit.RANDOM initialization

- Linear probing: intialized with average embedding (same as our method)

We summarize the number of trainable parameters for different baselines in the following table (base LM is LLaMA-7B). For our method, we assume using one domain marker and one functional token ($p = 10$), which is the case for all single-domain tasks in our evaluation, such as QED, Descriptor, and DC. For prompt tuning, there are two virtual prompts for consistency with our method, and the length of each learnable prompt is also 10.

| | Base Model | Ours | LoRA | Prompt Tuning | Linear Probing |
|---|---|---|---|---|---|
| # Trainable Params | 7B | 86016 | 4194304 | 81920 | 4096 |

### A.2.2 NEAREST NEIGHBOR BASELINE

We additionally implemented a nearest neighbor baseline for QED and Descriptor tasks. The basic idea is that, for each data point in the test set, we want to find the most similar data point in the training set, and use the label of this most-similar data as our prediction. Since the data features are protein sequences and SMILES strings, we use the python SequenceMatcher function to select the nearest neighbor. However, this metric does not account for the intrinsic structure of the proteins and chemicals and thus performs poorly compared to the other learning-based approaches.

### A.2.3 GENERATION SETUP

For both our method and the baseline, when evaluating with generation, we simply greedily decoded the most likely sequence, as there are limited gains from using beam search with beam size $B = 4$.

