# OpenReview forum: "Mark My Words: Repurposing LLMs for Specialized Domains via Ability Tokens"
_ICLR.cc/2024/Conference — Submitted to ICLR 2024_

### Official Review · Reviewer_bkwo · 2023-10-31

**Soundness:** 2 fair
**Presentation:** 3 good
**Contribution:** 2 fair
**Rating:** 5
**Confidence:** 4

**Summary:**

This work proposes a parameter-efficient finetuning approach similar to "prompt tuning", which inserts special tokens to the input and learns to adapt domains and shift behaviors based on the special tokens. Specifically, two types of tokens can be learned: "domain markers", which prepends to the input to indicate specific domains, and "functional tokens", which appends to the input to indicate specific tasks. The authors conduct experiments on: 1) Machine Translation, to show the proposed approach can achieve modularity and compositionality of different domains; 2) Molecular Property Prediction & Drug-Drug Interaction, where the approach achieves better performance than other baselines when adopting regression heads (rather than to predict discrete tokens); 3) Binding Affinity Prediction, where the approach operates on both Protein and Drug domains and achieves good performance.

**Strengths:**

- The proposed approach is an effective application of "prompt tuning" with certain adaptation; especially, it is proven effective achieving modularity and compositionality through experiments on Machine Translation and Drug-Drug Interaction. It is shown that modularizing domains and tasks is possible through learning those special tokens, which could derive zero-shot performance on unseen task domains by composition.

- The proposed approach is evaluated on multiple datasets across different domains, especially including Protein and Chemical Compounds, which are quite distant from natural languages, where it is able to obtain good performance on all of them through finetuning only a small amount of parameters.

**Weaknesses:**

- The proposed approach is similar to "prompt tuning" and its related techniques. The mere adaptation is to provide a different set of tokens per domain/task, which itself is relatively trivial. The ability to achieve modularity and compositionality is also not as surprising, though it is still valuable to show it empirically.

- The state-of-the-art performance on Protein and Chemical Compounds seems to mainly come from using regression heads, rather than from this specific way of using the prompt tuning, as shown by Figure 3. **There are no side-by-side experiments comparing the traditional prompt-tuning and the proposed tuning both adopting regression heads or regular LM heads**. In the end, the proposed approach is essentially equivalent to prompt tuning, if not considering task/domain composition. Adding this side-by-side experiments on a specific task could help to show the advantages of the proposed setting.

**Questions:**

Is it possible to show the results of the regular prompt tuning with regression heads on the task of Descriptor Prediction or QED Prediction alone?

---

> ### Author Response · Authors · 2023-11-21
>
> We appreciate your valuable feedback and constructive comments. We have incorporated them in the revised paper. Below, we answer the questions raised in the review.
>
> &nbsp;
>
> > **More ablations: _"There are no side-by-side experiments comparing the traditional prompt-tuning and the proposed tuning both adopting regression heads or regular LM heads.""Is it possible to show the results of the regular prompt tuning with regression heads on the task of Descriptor Prediction or QED Prediction alone?"_**
>
> Thank you for your suggestion. We added these baselines to our rebuttal (see Table 1 in the general comment above). Specifically, we evaluate PEFT baselines w. regression head and compare them to our method (ability tokens w. regression head). This setting removes the potential influence of the prediction head and the loss function to help us better understand the effect of ability tokens. The results show that our method outperforms prompt tuning on all tasks and outperforms all PEFT methods extended with the regression head on 3 of 4 tasks. The only exception is the binding affinity task, where our method underperforms LORA w. reg head, which nevertheless uses 12x more trainable parameters than we do (~1M vs. ~86K).
>
> Apart from the "w. regression head" baselines you suggest, we add another set of ablation studies in Table 2 of the general comment above. We replace the learned embeddings of the domain markers with the text embeddings of the domain names (e.g., inserting the actual tokens "<", "Protein", ">" into the input sentence) to examine whether the domain markers have indeed learned useful domain knowledge from the auxiliary data. Our method still achieves outstanding performance, so domain markers are effective and necessary for improving downstream performance in specialized domains.
>
> Beyond these new experiments, our original submission contains detailed ablation studies for different components of our approach (see Section 4.2.2, Figure 3, and Appendix A.1.3), showing the performance of
> - Full method: domain markers + functional tokens + prediction head
> - w/o domain markers
> - w/o functional tokens
> - w/o prediction head
>
> The first setting obtains the best results. We have also studied the effect of ability token length in Figure 3.
>
> &nbsp;
>
> > **Difference from prompt tuning: _"The proposed approach is similar to 'prompt tuning' and its related techniques. The mere adaptation is to provide a different set of tokens per domain/task, which itself is relatively trivial."_**
>
> We underscore our difference from existing PEFT methods (including prompt tuning) in Section 2.1 and 3.1. In particular, our method separates task domains from task functions—we use **modularized** tokens to represent different domains/functions and learn them using a **hierarchical** training protocol. These ideas are **not** trivial because they lead to distinct properties compared to existing methods:
> * **Reusability & generalization to unseen tasks**:  While existing PEFT methods require learning a distinct set of parameters for every single task, regardless of whether they share domains, our learned tokens can be shared across tasks and applied to new problems without additional training. For example, as demonstrated in Section 4.2, the <SMILES> marker can be independently applied to both drug combination and the binding affinity task. Additionally, in Section 4.1, the general functional token <Translate> exhibits zero-shot transferability to unseen tasks.
> * **Local applicability & multiple occurrence**: Prompt tuning is "global" (applies to the entire query) and therefore always appears at the beginning, whereas our markers are "local" (apply to certain substrings of the input) and therefore can appear anywhere and multiple times in the prompt.
> * **Taking advantage of auxiliary data**: For existing PEFT methods, learning the additional parameters (e.g., soft prompts, adapters) uses only the target dataset. In contrast, our design of domain markers allows us to leverage auxiliary unsupervised data and learn domain knowledge from them, which yields better empirical results than the PEFT baselines (Table 2 & 3).
> * **Sample efficiency for downstream tasks**: By inserting learned domain markers into the input, we amortize the sample complexity needed to learn the functional tokens and solve downstream tasks.
>
> Apart from these desirable properties, we also show in the new experiments (Table 1 of the general comment above) that our method significantly outperforms prompt tuning, which further provides empirical evidence that these methods differ not only conceptually, but also functionally.
>
> Thank you for your question. We will add the above discussion to the paper for better clarity.

---

### Official Review · Reviewer_VcX3 · 2023-11-01

**Soundness:** 3 good
**Presentation:** 2 fair
**Contribution:** 3 good
**Rating:** 5
**Confidence:** 4

**Summary:**

The authors design a framework for adding trainable special tokens (called ability tokens) to pre-trained language models (LMs). Embeddings of these special tokens are learned on the corpus of specialized domains in order to adapt the model to these domains. The authors introduce two types of ability tokens: domain markers and functional tokens. Domain markers are trained on the single-domain unlabeled corpus. Functional tokens are trained on single-domain and multi-domain labeled samples. During inference, these special tokens are inserted into the input text. Experimental results on machine translation, protein, and molecular property prediction achieve better performance compared with other domain adaption methods, such as LoRA and prompt tuning.

**Strengths:**

1. This method achieves better performance in the medical domain than other PEFT methods.
2. These ability tokens can be combined to generalize to unseen tasks.
3. Only a few parameters need to be trained for the domain adaption.

**Weaknesses:**

1. This method is similar to existing PEFT methods like Prompt-Tuning. I think it may lack novelties.
2. Ablation experiments show that the effectiveness of domain markers is relatively limited.
3. I think that training regression heads for numerical prediction problems may cause an unfair comparison with other adaption methods based on text generation.

**Questions:**

1. What are the main differences between your approach and Prompt Tuning? It would be helpful to add a discussion about this.
2. Do you use the task-specific linear heads for other baseline methods, like LoRA and prompt tuning?
3. What is the difference between ability tokens with text instructions used in chat-aligned LLMs?

---

> ### Author Response · Authors · 2023-11-21
> **Official Comment by Authors 1/2**
>
> Thank you for your feedback and constructive comments, which we have incorporated in the revised paper. Below, we answer the questions raised in the review.
>
> &nbsp;
>
> > **Difference from previous PEFT methods: _"This method is similar to existing PEFT methods like Prompt-Tuning…What are the main differences between your approach and Prompt Tuning? It would be helpful to add a discussion about this."_**
>
> We underscore our difference from existing PEFT methods (including prompt tuning) in Section 2.1 and 3.1. In particular, our method separates task domains from task functions—we use **modularized** tokens to represent different domains/functions and learn them using a **hierarchical** training protocol. This leads to:
> * **Reusability & generalization to unseen tasks**:  While existing PEFT methods require learning a distinct set of parameters for every single task, regardless of whether they share domains, our learned tokens can be shared across tasks and applied to new problems without additional training. For example, as demonstrated in Section 4.2, the <SMILES> marker can be independently applied to both drug combination and the binding affinity task. Additionally, in Section 4.1, the general functional token <Translate> exhibits zero-shot transferability to unseen tasks.
> * **Local applicability & multiple occurrence**: Prompt tuning is "global" (applies to the entire query) and therefore always appears at the beginning, whereas our markers are "local" (apply to certain substrings of the input) and therefore can appear anywhere and multiple times in the prompt.
> * **Taking advantage of auxiliary data**: For existing PEFT methods, learning the additional parameters (e.g., soft prompts, adapters) uses only the target dataset. In contrast, our design of domain markers allows us to leverage auxiliary unsupervised data and learn domain knowledge from them, which yields better empirical results than the PEFT baselines (Table 2 & 3).
> * **Sample efficiency for downstream tasks**: By inserting learned domain markers into the input, we amortize the sample complexity needed to learn the functional tokens and solve downstream tasks.
>
> Apart from these desirable properties, we also show in the new experiments (Table 1 of the general comment above) that our method significantly outperforms prompt tuning, which further provides empirical evidence that these methods differ not only conceptually but also functionally.
>
> Thank you for the suggestion. We will add the above discussion to the paper for better clarity.
>
> &nbsp;
>
> > **Difference from text instructions: _"What is the difference between ability tokens with text instructions used in chat-aligned LLMs?"_**
>
> The biggest difference is that **our ability tokens are not represented as explicit text or actual words**. Besides, our method has the following functional benefits:
> - **Knowledge compression**: Ability tokens can compress large amounts of domain knowledge (via domain markers) or task knowledge (via functional tokens) into a few embedding parameters in a data-driven fashion. Such information is hard to summarize in a few sentences as text instructions.
> - **Task-agnostic**: Ability tokens are more general, e.g., domain markers are task-agnostic and can be reused for new tasks in the same domain. Text instructions are always task-specific.
> - **Flexibility & generalization to unseen tasks**: Ability tokens are composable and can generalize to unseen tasks in a zero-shot fashion, as exemplified in our multilingual experiments (Section 4.1).
> - **Effectiveness**: Ability tokens are more effective and stable compared to hand-crafted text prompts, as shown in Table 2 & 3, where our method outperforms the hard prompt baseline by a large margin.
>
> &nbsp;
>
> > **More baselines: _"I think that training regression heads for numerical prediction problems may cause an unfair comparison with other adaption methods based on text generation.""Do you use the task-specific linear heads for other baseline methods, like LoRA and prompt tuning?"_**
>
> Thank you for your suggestion. We added these baselines to our rebuttal (see Table 1 in the general comment above). Specifically, we evaluate PEFT baselines w. regression head and compare them to our method (ability tokens w. regression head). This setting removes the potential influence of the prediction head and the loss function to help us better understand the effect of ability tokens. The results show that our method outperforms prompt tuning on all tasks and outperforms all PEFT methods extended with the regression head on 3 of 4 tasks. The only exception is the binding affinity task, where our method underperforms LORA w. reg head, which nevertheless uses 12x more trainable parameters than we do (~1M vs. ~86K).

---

> ### Author Response · Authors · 2023-11-21
> **Official Comment by Authors 2/2**
>
> > **More ablations: _"Ablation experiments show that the effectiveness of domain markers is relatively limited."_**
>
> Apart from the "w. regression head" baselines you suggest, we add another set of ablation studies in Table 2 of the general comment above. We replace the learned embeddings of the domain markers with the text embeddings of the domain names (e.g., inserting the actual tokens "<", "Protein", ">" into the input sentence) to examine whether the domain markers have indeed learned useful domain knowledge from the auxiliary data. Our method still achieves outstanding performance, so domain markers are effective and necessary for improving downstream performance in specialized domains.
>
> Besides these new experiments, our original submission contains detailed ablation studies for other components of our approach (see Section 4.2.2, Figure 3, and Appendix A.1.3), showing the performance of
> * Full method: domain markers + functional tokens + prediction head
> * w/o domain markers
> * w/o functional tokens
> * w/o prediction head
>
> The first setting obtains the best results. We have also studied the effect of ability token length in Figure 3.

---

### Official Review · Reviewer_4P5B · 2023-11-04

**Soundness:** 3 good
**Presentation:** 3 good
**Contribution:** 2 fair
**Rating:** 5
**Confidence:** 4

**Summary:**

This paper explores the repurposing of general language models (LMs) as specialized task solvers, particularly in domains that have limited representation in pre-training corpora. The authors propose the use of ability tokens, specifically domain markers and functional tokens, to enhance LMs' ability to handle specialized inputs. Domain knowledge is encoded in domain markers, while specific task knowledge is encoded in functional tokens. However, a major concern is that the performance seems to be primarily driven by overlap between the training data of the LLMs and the downstream tasks. Additionally, the experiments are mainly focused on one specific domain without testing the generalization of the approach to other domains.

**Strengths:**

- The paper presents a straightforward method to incorporate domain knowledge into LMs while maintaining the original knowledge encoded in the models' parameters.
- The performance of the approach is not sensitive to different context lengths.

**Weaknesses:**

- The contribution of data contamination to the final performance is unclear. The ability tokens, which generally correspond to dataset or task names, may result in LMs memorizing information about the datasets during pretraining. An analysis of this phenomenon is needed to understand the effectiveness of the method.
- The proposed method is specifically designed for effective domain adaptation of LLMs, but it is only evaluated in the biomedical domain. Evaluations in other domains would strengthen the paper's findings.
- More ablation studies are required to demonstrate the effectiveness of the ability tokens. For example, evaluating the llama-7b model with and without ability tokens for different tasks.
- The experiments only utilize one model. Including results from other models would further support the conclusions.

**Questions:**

- Is there any specific procedure for initializing the embedding for ability tokens, or is it done in a standard manner?
- Are the lengths of domain markers and functional tokens always the same, or can they vary depending on the task?

---

> ### Author Response · Authors · 2023-11-21
> **Official Comment by Authors 1/2**
>
> Thank you for your feedback and constructive comments, which we have incorporated in the revised paper. Below, we answer the questions raised in the review.
>
> &nbsp;
>
> > **Data contamination: _"However, a major concern is that the performance seems to be primarily driven by overlap between the training data of the LLMs and the downstream tasks.""The ability tokens, which generally correspond to dataset or task names, may result in LMs memorizing information about the datasets during pretraining."_**
>
> We would like to clarify a couple of points to address these misunderstandings about our work.
>
> **First, it is important to note that the ability tokens do not _"correspond to dataset or task names."_ They exist only in the embedding space and are not represented as actual words.** For example, to learn the token representing the SMILES domain, we introduce "<ability token 0>" to the tokenizer with a newly initialized embedding. Then, we prepend "<aibility token 0>" (not the word "SMILES" or any dataset name related to SMILES) to SMILES strings and learn the embedding for "<aibility token 0>" via next-token prediction. At inference time, "<ability token 0>" is inserted into the input. Similarly, when learning the binding affinity function, the functional token added to the tokenizer is "<aibility token 1>", and its embedding is not tied to the task name "binding affinity."
>
> **Second, the datasets used to train the ability tokens do not appear in pretraining.** Our goal of using ability tokens is to inject *additional* domain/task knowledge into the LLM by leveraging *auxiliary* data. For instance, the dataset we use to train the <Protein> marker is from a recently published source [1], which should not have appeared in the pretraining corpora.
>
> **Lastly, the datasets used for evaluation do not appear in pretraining.** For instance, we generate the QED and descriptor prediction datasets *ourselves* using specialized Python packages. These data have not appeared anywhere on the web, so there should not be a memorization problem.
>
> We hope this clarifies any misconceptions. Please feel free to share if you have additional concerns. Thank you!
>
> [1] Blanchard AE, Gounley J, Bhowmik D, et al. Language models for the prediction of SARS-CoV-2 inhibitors. The International Journal of High Performance Computing Applications. 2022; 36(5-6):587-602.
>
> &nbsp;
>
> > **More ablations: _"More ablation studies are required to demonstrate the effectiveness of the ability tokens. For example, evaluating the llama-7b model with and without ability tokens for different tasks."_**
>
> We have performed such ablations in the paper. In particular:
> * The "hard prompt" baseline in Table 2 and 3 is **llama-7b without ability tokens and regression head**. This demonstrates the effect of using both ability tokens and regression head.
> * The "w/o marker" baseline in Figure 3 (and Appendix A.1.3) is llama-7b with functional tokens and regression head but **without domain markers**. This demonstrates the effect of domain markers.
> * The "w/o functional" baseline in Figure 3 (and Appendix A.1.3) is llama-7b with domain markers and regression head but **without functional tokens**. This demonstrates the effect of functional tokens.
>
> For rebuttal, we add two additional sets of ablations as requested (see the general comment above).
> * In Table 1, we evaluate PEFT baselines w. regression head and compare them to our method (ability tokens w. regression head). This setting removes the potential influence of the prediction head and the loss function to help us better understand the effect of ability tokens. The results show that our method outperforms prompt tuning on all tasks and outperforms all PEFT methods extended with the regression head on 3 of 4 tasks. The only exception is the binding affinity task, where our method underperforms LORA w. reg head, which nevertheless uses 12x more trainable parameters than we do (~1M vs. ~86K).
> * In Table 2, we replace the learned embeddings of the domain markers with the text embeddings of the domain names (e.g., inserting the actual tokens "<", "Protein", ">" into the input sentence) to examine whether the domain markers have indeed learned useful domain knowledge from the auxiliary data. Our method still achieves outstanding performance, so domain markers are effective and necessary for improving downstream performance in specialized domains.

---

> ### Author Response · Authors · 2023-11-21
> **Official Comment by Authors 2/2**
>
> > **More domains: _"Additionally, the experiments are mainly focused on one specific domain without testing the generalization of the approach to other domains.""Evaluations in other domains would strengthen the paper's findings."_**
>
> Thank you for the suggestion. We showcase the general effectiveness of our approach across a diverse set of ten domains, encompassing eight languages, protein sequences ("MTVPDRSEIAGKWYVV…"), and chemical formulas ("CCO[C@H](C(=O)O)Cc1…").  We deliberately chose the latter two domains because (1) they represent non-linguistic, highly specialized domains with distributions significantly divergent from natural language distributions; (2) they are relatively well-understood with established baselines that we can compare to; and (3) they are substantially different from each other, allowing us to investigate the interactions between them. We acknowledge the potential for expanding the scope of domains in future iterations of this work.
>
> &nbsp;
>
> > **Embedder initialization: _"Is there any specific procedure for initializing the embedding for ability tokens, or is it done in a standard manner?"_**
>
> We have outlined the initialization procedure in Section 3.2 (last paragraph on page 4). Specifically, the ability tokens are initialized using the average embedding of the LLM's original dictionary, scaled by a factor to match the norm of this average embedding with the average norm of all token embeddings in the LLM's original dictionary.
>
> &nbsp;
>
> > **Ability token length: _"Are the lengths of domain markers and functional tokens always the same, or can they vary depending on the task?"_**
>
> The length $p$ is specified as a hyperparameter in Section 3.2 (see the 6th-to-last line on page 4), so it is variable. We explore its impact on downstream performance using the drug combination dataset  (Section 4.2.2 and Figure 3).

---

### Official Review · Reviewer_gWMh · 2023-11-11

**Soundness:** 4 excellent
**Presentation:** 3 good
**Contribution:** 3 good
**Rating:** 6
**Confidence:** 3

**Summary:**

This paper discusses the limitations of Large Language Models (LLMs) in highly specialized fields such as biomedical sciences and introduces a new framework to improve their performance in specialized tasks. The authors propose the use of "ability tokens" as domain markers to guide the model in specific tasks.

The setting of this paper is interesting. While adding ability tokens is a common practice in aligning language models with new tasks and modalities, it is non-trivial as it results in changes in the early layers of LLM embeddings. The three-stage hierarchical training protocol is both novel and practical. Additionally, the authors evaluate the method across a wide spectrum of tasks, further demonstrating its applicability.

**Strengths:**

1. The paper is well-written, and the claims of the paper are supported by comprehensive experiments on a wide spectrum of tasks. The drug discovery experiments, in particular, are solid, given that biosequences deviate significantly from text, thereby qualifying as a distinctive modality.
2. The method has good generalization properties (could generalize to unseen tasks).
3. The study introduces a novel approach that provides contribution to the field, particularly when compared to traditional prompt-tuning methods. The hierarchical training of ability tokens as proposed in the paper enhances the potential for generalization across varied tasks, and allows the combination of ability tokens, making parameter-efficient methods more suitable for multi-task learning.

**Weaknesses:**

1. The paper lacks ablation results, which is crucial to demonstrate the effectiveness of the ability tokens.

**Questions:**

1. What is the effectiveness of the 3-stage training process and how much each stage contributed to generalization results?
2. A common practice in adapting Llama to specialized domains is to add specialized tokens as words, such as <molecule></molecule>.  However, these tokens are not added as new tokens, but are tokenized into multiple tokens '<', 'molecule', '>',  and then use LORA to adapt the llm to specialized tasks. This method has shown effective in [1][2]. What is the advantage of the proposed methods to this way of adding ability tokens?
3. What is the data efficiency of the proposed method, compared to prompt tuning and LORA?


[1] Zhu, Deyao, et al. "Minigpt-4: Enhancing vision-language understanding with advanced large language models." arXiv preprint arXiv:2304.10592 (2023).
[2] Liu, Haotian, et al. "Visual instruction tuning." arXiv preprint arXiv:2304.08485 (2023).

---

> ### Author Response · Authors · 2023-11-21
> **Official Comment by Authors 1/2**
>
> We appreciate your valuable feedback and constructive comments. We have incorporated them in the revised paper. Below, we answer the questions raised in the review.
>
> &nbsp;
>
> > **More ablations: _"The paper lacks ablation results, which is crucial to demonstrate the effectiveness of the ability tokens."_**
>
> Thank you for your suggestion. We conducted two additional sets of ablation experiments to provide further understanding of the contribution of the ability tokens. The detailed results are shown in the general comment above.
> * In Table 1, we evaluate PEFT baselines w. regression head and compare them to our method (ability tokens w. regression head). This setting removes the potential influence of the prediction head and the loss function to help us better understand the effect of ability tokens. The results show that our method outperforms prompt tuning on all tasks and outperforms all PEFT methods extended with the regression head on 3 of 4 tasks. The only exception is the binding affinity task, where our method slightly underperforms LORA w. reg head, which nevertheless uses 12x more trainable parameters than we do (~1M vs. ~86K).
> * In Table 2, we replace the learned embeddings of the domain markers with the text embeddings of the domain names and prompt the LLM, as you suggest. Our method still achieves outstanding performance, which indicates that domain markers have indeed learned useful domain knowledge from the auxiliary unsupervised data.
>
> Beyond these new experiments, our original submission contains detailed ablation studies for different components of our approach (see Section 4.2.2, Figure 3, and Appendix A.1.3), showing the performance of
> * Full method: domain markers + functional tokens + prediction head
> * w/o domain markers
> * w/o functional tokens
> * w/o prediction head
>
> The first setting obtains the best results. We have also studied the effect of ability token length in Figure 3.
>
> &nbsp;
>
> > **Understanding training protocol: _"What is the effectiveness of the 3-stage training process and how much each stage contributed to generalization results?"_**
>
> In each stage of the 3-stage training process, we learn a different type of ability token. The effect and contribution of individual stages are summarized below:
>
> * **Stage 1: training domain markers.** This stage allows us to inject general domain knowledge into the prompting process. It improves downstream performance since the "w. domain markers" setting outperforms the "w/o domain markers" setting (Figure 3). This stage also lays out the foundation for later stages because training functional tokens requires learned domain markers. *It contributes the most to the generalization results, given that domain markers are task-agnostic and applicable to various in-domain tasks.*
> * **Stage 2: training single-domain functional tokens.** This stage allows the model to learn task knowledge solely from feature-label pairs (without human instructions) and compress the information into a few embedding parameters. Stage 2 boosts the efficacy of our method since the "w. functional tokens" setting outperforms the "w/o functional tokens" setting (Appendix A.1.3).
> * **Stage 3: training multi-domain functional tokens.** This stage is similar to stage 2, but the target functions are multi-domain, so we extend the capacity of LLMs to solving multi-domain tasks like drug-protein binding affinity prediction (Section 4.2.3). Moreover, as we show in the multilingual translation experiments (Section 4.1), multi-domain tokens like <Translate> can be combined with different markers and applied to unseen tasks during training. *Thus, this stage also contributes to our markup system's zero-shot generalization ability.*
>
> We will add the above discussion to the paper for better clarity.

---

> ### Author Response · Authors · 2023-11-21
> **Official Comment by Authors 2/2**
>
> > **Relevant work: _"A common practice in adapting Llama to specialized domains is to add specialized tokens as words… What is the advantage of the proposed methods to this way of adding ability tokens?"_**
>
> Thank you for pointing out this line of related work. The approach you referenced involves two key components: adding "<", "molecule", ">" to the input, and using adapters like LORA. We discuss our advantages relative to both components:
> * **Better controllability & conditioning effect.** The efficacy of using pretrained tokens like "<", "molecule", ">" relies heavily on their occurrence in the pretraining corpora. However, end users do not have control over this, e.g., we are uncertain how frequently the word “molecule” occurs (if the frequency is low, then adding the tokens may not be useful) and whether these occurrences align with the target domain. For example, the domain of "molecule" is different in the sentence *"2 atoms of H and 1 of oxygen form a molecule of water"* (chemistry) and the sentence *"the four molecules of life are proteins, carbohydrates, lipids and nucleic acids"* (biology). Our method addresses this limitation by learning continuous embeddings for specialized domains *explicitly using data from the target domain*. This allows more fine-grained control over how we condition the LLM.
> * **Generalization to unseen tasks.** Using LORA after adding "<", "molecule", ">" to the input still necessitates *learning a distinct set of parameters for each downstream task*, even though the tasks may share knowledge, as discussed in Section 3.1. On the contrary, our learned domain markers can be used for various tasks in the same domain (e.g., the <SMILES> token can be applied to both drug combination and binding affinity prediction). The learned functional tokens also exhibit zero-shot generalization to unseen domains (e.g., the <Translate> token can generalize to unseen language pairs, as we show in Section 4.1). Thus, our ability tokens are reusable, lowering the deployment costs for new tasks.
> * **Taking advantage of auxiliary data**: For existing PEFT methods, learning the additional parameters (e.g., soft prompts, adapters) uses only the target dataset, whereas our design of domain markers leverages auxiliary unsupervised data and learns domain knowledge from them. This yields better empirical results (Table 2 & 3) and amortizes the sample complexity needed to learn the functional tokens.
>
> To provide a more direct comparison between our method and the approach you mention, we evaluate LORA/prompt tuning with domain names inserted into the input text (see Table 2 in the general comment). The results show that our method is more effective in addition to being more controllable and generalizable.
>
> Thank you for the references. We will add the above discussion to the paper.
>
> &nbsp;
>
> > **Data efficiency: _"What is the data efficiency of the proposed method, compared to prompt tuning and LORA?"_**
>
> We discuss data efficiency in two scenarios:
> * **Case 1: no relevant domain marker or functional token learned yet.** Prompt tuning and LORA learn task-specific parameters entirely from the target dataset. However, our method requires learning task-agnostic domain markers using unsupervised domain data, in addition to learning the functional token using the target dataset. This entails using more data than the PEFT methods, but in practice, the additional unsupervised data is more widely available and much easier to obtain (or generate) than labeled data.
> * **Case 2: there are relevant markers and functional tokens trained for previous tasks which we can use off-the-shelf.** For instance, in our multilingual translation experiments, we have already learned various language markers and the functional token. Thus, we can generalize zero-shot to new translation tasks by directly inserting the learned ability tokens into the input. As we don't have to train any additional parameters, we achieve better data efficiency than the PEFT methods, which still require learning new parameters for new tasks.
>
> We will add the above discussion to the paper.

---

### Author Response · Authors · 2023-11-21
**General comment regarding new baselines and ablation studies**

We appreciate the reviewers' constructive feedback and comments. We are encouraged by their appreciation of the strengths of the paper, particularly our method's generalization (gWMH), modularity (VcX3, bkwo), and robustness (4P5B) properties; the comprehensiveness of the experiments (gWMH), and the good empirical performance relative to comparable state-of-the-art methods (VcX3).

To address shared concerns expressed by the reviewers and to complement the empirical results in our original submission, **we added two sets of experiments in the rebuttal, showing that:**
* The proposed approach of learning and using ability tokens significantly improves the LLM’s performance on specialized tasks over naive prompt tuning and is much more parameter-efficient than adapter-based methods like LORA.
* Domain markers can extract useful domain knowledge from auxiliary data, compress them into a few embedding parameters, and leverage them effectively for task-solving.

&nbsp;

### **Ablation of Prediction Head and Loss Function**

**Table 1:** We evaluate PEFT baselines with regression head (trained using MSE loss) to better understand the effect of ability tokens, since this setting removes the potential influence of the prediction head and the loss function:

|| # Trainable Params|Descriptor prediction (MSE) |QED prediction (MSE)|  Drug combination (MAE) |  Binding affinity (Pearson $r$)|
|:-:|:-:|:-:|:-:|:-:|:-:|
|LORA w. reg head |1052672 | 0.007  |0.015  | 12.53 |**0.534**|
|Prompt tuning w. reg head | 86016 | 0.008 | 0.0083 |15.29 | 0.38 |
|Linear probing (reg head only)  |4096|0.041  |0.012  | 24.11 | 0.18 |
|**Ours (ability tokens w. reg head)** |86016 |**0.005** |**0.008** | **12.21** | 0.527 |

On all 4 tasks, our method outperforms naive prompt tuning, showing the effect of injecting domain knowledge into the LLM via domain markers. On 3 of 4 tasks, we outperform all PEFT w. regression head baselines. This indicates that our way of learning and using ability tokens is generally effective and necessary for improving downstream performance. The only exception is the binding affinity task, where we underperform LORA w. reg head only by 0.007, but LORA uses 12x more trainable parameters than we do (1052672 vs. 86016).

&nbsp;

### **Ablation of Domain Markers**

**Table 2:** We replace the learned embeddings of the domain markers with the text embeddings of the domain names (e.g., inserting the actual tokens "<", "Protein", ">" into the input sentence) and prompt the LLM. The goal is to verify whether the domain markers have indeed learned useful domain knowledge from the auxiliary unsupervised data. Note that regression head and MSE loss are used in this setting.

|| # Trainable Params|Descriptor prediction (MSE) |QED prediction (MSE)|  Drug combination (MAE) |  Binding affinity (Pearson $r$)|
|:-:|:-:|:-:|:-:|:-:|:-:|
|LORA w. domains (text)|1052672 | 0.008 |  0.015 | 12.79 |**0.552**|
|Prompt tuning w. domains (text) | 86016 | 0.007 | 0.0085 | 14.73| 0.37 |
|Linear probing w. domains (text)|4096|0.049  |0.011  | 14.64 | 0.31 |
|**Ours (learned domain markers)** |86016 |**0.005** |**0.008** | **12.21** | 0.527 |

Compared to the first table, simply adding the domain as text to the input brings marginal benefits (or sometimes negative effects) to prompt tuning and LORA. On all tasks, we outperform prompt tuning, so our approach effectively compresses domain knowledge and leverages auxiliary data to help solve downstream tasks. The gap between our method and LORA on binding affinity is again due to the fact that we use much fewer trainable parameters.

&nbsp;

We will add the above results to the revised version of our paper.

---

### Meta-Review · Area_Chair_7YHv · 2023-12-11

**Metareview:**

This paper propose the use of ability tokens, specifically domain markers and functional tokens, to enhance LMs' ability to handle specialized inputs. Domain knowledge is encoded in domain markers, while functional tokens guides the model on specific tasks. The results show improvement on molecular property prediction, zero-shot generational and is a promising way to improve specific task performance while maintaining general capabilities.

Reviewers liked the simplicity of the method, the zero-shot generalization, modularity, and empirical strength. On the novelty side, several reviewers felt this is a simple adaptation from prompt-tuning. On the experiment side, reviewers objected to the limited ablations, domains and models tested. The authors included more baselines and ablation comparing the proposed method to LORA with mostly (3/4) favorable results, but did not include more domains or models. The authors also made arguments on the difference of their method with prompt tuning and why it should count as a major improvement over prompt-tuning. Judging this seems to be a matter of opinion and not a fundamental misunderstandings, and the reviewers provided their opinions.  Unfortunately the reviewers did not respond to these improvements and the AC did not encourage more discussions.

**Justification For Why Not Higher Score:**

reviewer majority, no fundamental misunderstanding, no one favored strong accept

**Justification For Why Not Lower Score:**

N/A

---

### Decision · Program_Chairs · 2024-01-16

Reject